# Domestic Cattle in a National Park Restricting the Sika Deer Due to Diet Overlap

**DOI:** 10.3390/ani13040561

**Published:** 2023-02-05

**Authors:** Feng Wu, Di Zhu, Peiying Wen, Zhizhen Tang, Lei Bao, Yu Guan, Jianping Ge, Hongfang Wang

**Affiliations:** 1National Forestry and Grassland Administration Key Laboratory for Conservation Ecology in the Northeast Tiger and Leopard National Park, Beijing 100875, China; 2Northeast Tiger and Leopard Biodiversity National Observation and Research Station, Beijing 100875, China; 3Ministry of Education Key Laboratory for Biodiversity and Ecological Engineering, College of Life Sciences, Beijing Normal University, Beijing 100875, China

**Keywords:** DNA barcoding, foraging niche partition, dietary overlap, cattle diet shift, sika deer

## Abstract

**Simple Summary:**

Precisely estimating the extent of the conflict between livestock and wild animals is crucial for making a proper management policy. Sika deer in the Northeast Tiger and Leopard National Park restrictedly distributes along the Sino-Russia border and fail to dispersal further into inland China, which hampers the ecosystem restoration flagged with tiger and leopard. Domestic cattle raised in the park are hypothesized as the main factors restricting the dispersal of sika deer, while the extent of the cattle-deer conflict is largely unknown. By utilizing high-throughput sequencing technology, this study quantified the extent of foraging conflict between cattle and deer in the park. We found the cattle shifted from a grazer-typical diet to a browser-typical diet, which resulted in a big diet overlap with sika deer. Indeed, the diet of cattle was more diverse than that of sika deer. Therefore, we argue that the big diet overlaps and superior competitive abilities of cattle may be the main driver restricting the dispersal of sika deer into the inland.

**Abstract:**

Managers need to know the extent of the conflict between livestock and wild animals. Although many studies have reported the conflict between livestock and wild animals, few have checked the extent of the conflict. Cattle raising in the Northeast Tiger and Leopard National Park is considered one of the main driving forces behind the restricted distribution of sika deer. To understand whether foraging competition is contributing to avoidance patterns between sika deer and cattle, we investigated their feeding habits using DNA barcoding and high-throughput sequencing. Our study shows that although cattle are grazers in the traditional division of herbivores, their diet shifted to a predominance of dicotyledonous woody plants, and this diet shift resulted in a high degree of dietary overlap between sika deer and cattle. Moreover, compared to sika deer, cattle diets are more diverse at the species level with a wider ecological niche. Our results confirm that overlapping dietary niches and the superior competitive abilities of cattle contribute to the restricted distribution of the sika deer, which has critical implications for the conservation of their predators. Our study suggests that cattle grazing should be prohibited in the Park and effective measures should be taken for the benefit of sika deer.

## 1. Introduction

Human–wildlife conflict is an important issue in wildlife conservation and management. Grazing livestock in wildlife areas is becoming more common due to the increasing human demand for food [1,2]. This large-scale introduction of species can eventually transform existing ecosystems into agroecosystems, which may lead to a loss of species and functional diversity [3,4]. The coexistence of wild and domestic animals is common, especially in densely populated areas. Understanding the effects of domestic animals on wildlife and quantifying the conflict between the two is crucial to assessing the impact of human disturbance on wildlife population development and maintenance [5]. However, most studies have focused on the predation of livestock by wild carnivores or crop consumption by wild herbivores, while interactions and conflicts between wild and domestic herbivores have been largely overlooked [5,6,7,8].

The Northeast Tiger and Leopard National Park is a national park for the conservation of rare species of wild animals, such as the Amur tiger (Panthera tigris altaica) and Amur leopard (Panthera pardus orientalis). Sika deer (Cervus nippon) are the main large herbivorous mammals in the national park and the main prey of Amur tigers and leopards. The national park is characterized by cool temperate coniferous-broadleaved mixed forest, which is located in the northeast of China and contains large areas bordering on human living areas. According to the monitoring data from the camera trap network, sika deer in the park are restricted within the border with Russia, the distribution of which is the same as the tiger and leopard (Figure 1) [9]. Determining the reasons why sika deer are confined to the border area is critical for the recovery of deer populations and habitats, as well as the restoration of the entire national park ecosystem.

Cattle raising in the national park is considered one of the main driving forces behind the restricted distribution of sika deer [9,10] based on the spatial and temporal distribution patterns of wild and domestic herbivores in the national park using camera trap data. Sika deer are sensitive to the presence of cattle, as evidenced by decreased habitat use, spatial avoidance, and changing daily activity patterns. It is common for wild herbivores to respond to the presence of domesticated herbivores by decreasing habitat use, practicing spatial avoidance, and changing daily activity patterns [11,12,13,14]. However, the mechanism behind the spatial-temporal pattern of sika deer and cattle is still unknown. 

Domestic herbivores can affect wild herbivores in a number of ways, such as competition for food resources [14,15,16,17,18], the spread of diseases and parasites [19,20], fencing that limits the movement of wildlife [21,22,23], and the survival of deer newborns by reducing hiding places, as newborns are particularly sensitive to changes in vegetation cover [24,25]. Livestock grazing can alter the dietary niche of wild herbivores [26,27] and reduce forage quantity and quality, which in turn can limit the biomass of wild herbivore populations [28,29,30]. Furthermore, extensive dietary niche overlap under conditions of limited availability of resources could lead to competition, which may result in competitive exclusion and local extinctions [31]. Theoretically, cattle are classified as typical grazers in the traditional classification of herbivores, mainly feeding on monocotyledons. The native sika deer is an intermediate herbivore, feeding mostly on dicotyledons [32]. Direct competition for food resources between cattle and sika deer should be low. However, cattle are an introduced species, and their introduction into temperate forests may lead to changes in their feeding habits, resulting in foraging niche overlap with sika deer [33,34]. Competition for food resources may have contributed to the current distribution-limited spatial pattern of the relatively less competitive sika deer. Understanding the mechanism behind the spatial pattern is critical since it is the first step for effective livestock management [5].

In this study, we aim to understand whether foraging niche competition contributes to the avoidance pattern between sika deer and cattle. We used a recently developed sequencing-based, a next-generation DNA metabarcoding technique that enables highly sensitive, accurate, quantitative, and time-saving diet analysis of virtually any food type at a fine taxonomic resolution [35,36,37], which can reveal essential information about the interspecific competition and resource allocation that may be obscured in morphological diet analysis [38,39,40]. Multiple perspectives such as species level and phylogenetic level were used to analyze the feeding characteristics of the two species, the degree of feeding overlap, and to discuss direct conflicts in feeding and possible indirect effects through the plant community.

## 2. Materials and Methods

### 2.1. Study Areas and Sampling

Our study area is located in the eastern part of the national park and spans an increasing gradient of human disturbance and land use and a decreasing gradient of ecosystem integrity from east to west. It has a mountainous landscape and rugged terrain, with elevations varying from 5 to 1477 m. The dominant vegetation types include Korean pine (*Pinus koraiensis*) forests, deciduous birch and oak forests, coniferous forests, natural shrublands, and agricultural areas. Ungulate species that may fall prey to tiger and leopard include the Siberian roe deer (*Capreolus pygarus*), the sika deer (*Cervus nippon*), and the wild boar (*Sus scrofa*).

Cattle grazing occurs in the study area only between May and October. Grazing is concentrated in the summer because spring and autumn are brief. Therefore, we chose to sample in August (2020 and 2021), the main growing season for local plants. Since the availability of different orders of plants may drive diet differentiation even within the same species [33], to avoid this potential sample bias, we include spatially proximate samples of sika deer and cattle. In detail, two cattle grazing places in the forest were located to collect cow feces, and the feces of sika deer were collected within five kilometers of the grazing places. We did not investigate the vegetation around feces samples, but assumed that the food availability was comparable between samples of the two species according to our sample strategy. When multiple feces were present in close proximity, only one sample was collected to avoid repeated sampling of the same feces. We selected relatively fresh samples for identification based on their moist surfaces to optimize the chances of success for molecular analysis. Samples were handled using sterile disposable gloves and put into an ice box for temporary storage, after which they were stored at −80 °C upon transfer to the laboratory at Beijing Normal University (Beijing, China).

### 2.2. DNA Extraction, PCR Amplification, and Sequencing

DNA was extracted from individual fecal samples in small batches (typically 12 samples) in a room designated for processing fecal samples. Before and after each batch, the table tops and associated equipment were treated with 75% ethanol and DNA AWAY. In each extraction batch, extraction blanks (containing no fecal samples) were processed to monitor contamination during extraction. The QIAamp Stool Mini Kit (Qiagen) was used to extract DNA from each 200 mg fecal sample. A total of seven negative controls were prepared during the DNA extraction process. 

A commonly used mini-barcoding fragment, psbCL, has been shown to have high taxonomic coverage and discriminative capacity for vascular plants [41]. We amplified this fragment with the primer pair psbCL_F (5′-TGGTTATTTACTAAAATC-3′) and psbCL_R (5′-TTTGGTTAAGATATGCCA-3′), the product of which was approximately 100 bp [42].

In order to distinguish samples from post-sequencing bioinformatics analysis, a 7-bp tag was added to the 5′ end of each forward and reverse primer. There are at least three base differences between the tags [43], which ensures that the PCR products of each tag represent unique individual samples. In total, there were 86 DNA samples, including 47 sika deer feces samples, 32 cattle samples, and seven negative controls.

All PCR amplifications were conducted in a total volume of 30 μL, including 15 μL (2×) GS Taq PCR Mix, 1.2 μL psbCL_F/R (10 μM), 8.6 μL ddH2O, and 4 μL template DNA. The PCR program was as follows: an initial denaturation step of 5 min at 95 °C, followed by 35 cycles of 30 s at 95 °C, 30 s at 45 °C, and 30 s at 72 °C, and a final cycle of 10 min at 72 °C. PCR products were checked by electrophoresis on a 1.3% agarose gel. Only PCR samples with clearly visible bands, except for the negative controls, were able to proceed to the next step. 

To improve the concentration of PCR products and reduce random amplification during PCR, three PCR replicates were performed independently for each DNA sample. After PCR, three PCR replicates were admixed and purified using the EasyPure^®^ PCR Purification Kit (TransGen). 

All purified PCR products from all 86 samples were mixed together by adding 10 μL of each product. High-throughput sequencing was performed at Tiangen Biochemical Technology and Tsingke Biotechnology by illumination with the Solexa NovaSeq 6000 (Illumina). 

### 2.3. Sequence Analysis and Filtering 

OBITools was used for the analysis of high-throughput sequencing results [44]. The direct and reverse sequences were aligned using the illumina-paired-end program. Aligned sequences with a quality score < 40 were removed using the obigrep command. The ngsfilter command was used to identify sequences with perfectly matched tags and primers and retained them for further analysis. Identical sequences were clustered into unique sequences using the obiuniq command. The obigrep command was used to remove sequences that were less than 80 bp or had a total count of less than 10 across the entire dataset. PCR and sequencing errors were detected and discarded using the obiclean command. To reduce the effect of contamination, samples with lower read counts than negative controls were removed.

Deagle et al. [37] suggested removing sequences with less than 1% read count to reduce field and laboratory contamination. Hence, we conducted the following analysis by removing sequences less than 1% read counts (hereafter “1% threshold”). However, the threshold chosen was arbitrary, and some true dietary taxa may have been lost during the contamination-removing procedure, which may sometimes lead to a totally different result [45]. To check whether thresholds influence the diet pattern, five thresholds (0.01%, 0.05%, 0.1%, 0.5%, and 1%) were tested, and the dietary pattern analysis was repeated for each species under each threshold.

Taxonomic identification of the clean reads was performed with BLAST using the NCBI nucleotide database. We used the 100% criteria to match the query sequence during the taxonomic identification. Sequences that failed to match with 100% identity within the reference database were recorded as unknown and removed from the subsequent analysis. Food composition was expressed at the level of plant family and the level of molecular operational taxonomic units (MOTUs) [46]. To further avoid bias during sequencing and data processing, we selected two sequencing companies for sequencing and calculated the effects of different thresholds during data processing.

### 2.4. Dietary Pattern of Cattle and Sika Deer

Dietary data were summarized across samples using a metric commonly used in molecular dietary data analysis: relative read abundance (RRA), a quantitative (sequence abundance-based) metric [37]. The RRA of each MOTU in each ungulate’s diet was calculated as:$$RRA_{i}=(\frac{1}{S}\overset{s}{\underset{k=1}{\sum }}\frac{n_{jk}}{\sum_{i=1}^{T}n_{jk}})\times 100\%$$
where *S* is the number of samples, *T* is the number of MOTUs, *n_i,k_* is the number of sequence reads of MOTU *i* in sample *k*. For each sample, RRA represented the mean values across the replicate PCRs. The RRA for each ungulate species represented the mean values across all relevant samples. Dietary analysis based on RRA data is less sensitive to the impacts of low-abundance foods, contamination, and the count threshold setting used in the sequencing data processing. 

Dietary niche breadth was evaluated using the taxonomic diversity and phylogenetic diversity of the plant taxa for individual species. Diet taxonomic diversity was estimated by Hill numbers of order q = 0, 1, and 2 (i.e., ^0^*D*, ^1^*D*, and ^2^*D*), Levins’ measure of niche breadth (*B_A_*) [47], and Peilou’s evenness (*J*) using the diversity function of the R package vegan. Hill numbers provide a unified framework for biodiversity measurement [45]: ^0^*D* equals species richness (the number of MOTUs); ^1^*D* measures species diversity weighted by their relative frequency (RRA) in the diet (the exponential of the Shannon index); and ^2^*D* equals the inverse of the Simpson index, which gives more weight to dominant species and thus is reflective of the evenness of the diet.

We used the two most commonly used metrics, the mean pairwise phylogenetic distance (MPD) and mean nearest taxon distance (MNTD) to estimate the phylogenetic diversity of diet. MPD reflects the overall phylogenetic clustering of taxa on a tree, while MNTD reflects the extent of terminal clustering regardless of deep clustering [48]. We estimated the phylogenetic relationships between plant MOTUs using Bayesian inference methods.

### 2.5. Interspecific Dietary Differences and Overlaps

Differences in the RRA-based dietary composition of two ungulate species were assessed by permutational analysis of variance (PERMANOVA) using the “Adonis” function with 999 permutations and visualized by non-metric multidimensional scaling (NMDS) using Bray–Curtis dissimilarity in the R package vegan.

The dietary overlap between cattle and sika deer was estimated first using Pianka’s index (*O_jk_*) [49] in the R package EcoSimR 0.1.0. Subsequently, a simulation matrix of 10,000 random diets was generated to determine if the ecological niche overlap was greater than the degree of overlap expected by chance. *O_jk_* was calculated as follows:$$O_{jk}=\frac{\sum_{i}^{n}p_{ij}p_{ik}}{\sqrt{\sum_{i}^{n}p_{ij}^{2}\sum_{i}^{n}p_{ik}^{2}}}$$
where *O_jk_* is Pianka’s measure of niche overlap between species *j* and *k*, *p_ij_* is the proportion (RRA) of the *i*th MOTU of the total used by species *j*, *p_ik_* is the proportion of the *i*th MOTU of the total used by species *k*, and *n* is the total number of MOTUs.

Dietary overlap between pairs of cattle and sika deer was also measured by Bray–Curtis similarity *(BCsim*). We first estimated the Bray–Curtis dissimilarity in vegan as follows
$$BC_{jk}=\overset{n}{\underset{i}{\sum }}\frac{\left|p_{ij}-p_{ik}\right|}{\left(p_{ij}+p_{ik}\right)}$$
where *BC_jk_* is the Bray–Curtis dissimilarity between the diet of species *j* and *k*, *p_ij_* is the proportion (RRA) of the *i*th MOTU of the total used by species *j*, and *p_ik_* is the proportion of the *i*th MOTU of the total used by species *k*. Next, the Bray–Curtis dissimilarity was subtracted from 1 to obtain *Bcsim*. Both *O_jk_* and *Bcsim* range from 0 to 1, with 0 indicating no dietary overlap and 1 indicating complete overlap.

## 3. Results

### 3.1. Sample and Sequence Summary

Using the psbCL region of chloroplasts for metabarcoding, we analyzed the main food composition of cattle and sika deer (total 79 samples, Sika deer: 47; Cattle: 32). We conducted metabarcoding in two companies and obtained 13,422,242 raw sequence reads from Tiangen and 10,377,258 raw sequence reads from Tsingke, respectively (Table 1). After quality filtering and bioinformatics processing, the results from the two companies contained 72 samples (Sika deer: 44; Cattle: 28) from Tiangen and 65 samples (Sika deer: 41; Cattle: 24) from Tsingke, respectively. Seven samples from Tiangen and fourteen samples from Tsingke were removed due to lower read counts than the negative. We compared the sequences with the NCBI database, combined with the survey data of native plants, and conducted a subsequent analysis based on the results of family-level identification, thus identifying 34 MOTUs from Tiangen and 30 MOTUs from Tsingke.

### 3.2. Dietary Pattern of Sika Deer

The results from Tiangen show that sika deer mainly feed on dicotyledons of Rosaceae (29.8%), Betulaceae (20.1%), Sapindaceae (14.1%), Urticaceae (11.1%), and Fagaceae (7.9%). The results for the Tsingke are essentially the same as those for the Tiangen, but the percentages of each family differ, respectively, as Rosaceae (30.8%), Betulaceae (20.5%), Sapindaceae (13.9%), Urticaceae (11.7%), and Fagaceae (8.9%). The results of the two companies on the diversity of the Sika deer differed slightly, with Tsingke’s diversity indicators generally lower than those of Tiangen, but none of the differences were significant (Table 2). The effects of different thresholds on the feeding habits of sika deer were analyzed using Tiangen results as an example. With the increase of threshold, both the Hill number value and niche width of sika deer decreased, while Peilou’s evenness gradually increased (Figure 2).

### 3.3. Dietary Pattern of Cattle

The results of Tiangen show that cattle mainly feed on monocotyledons and dicotyledons of Sapindaceae (20.2%), Urticaceae (18.9%), Rosaceae (15.1%), Betulaceae (14.8), and Cyperaceae (9.3%) (Figure 3A). The results of Tsingke showed that cattle mainly feed on dicotyledons of Sapindaceae (23.2%), Urticaceae (21.0%), Rosaceae (17.6%), Betulaceae (13.4%), and Rhinoceros/Labiaceae (6.3%) (Figure 3B). As for the cattle, the taxonomic diversity and threshold results for both companies have a similar pattern to that of the sika deer. Results from both companies showed significant differences in ^1^*D*, ^2^*D,* and Pielou’s evenness between cattle and sika deer, but no significant differences in ^0^*D*, and almost all diversity indicators of cattle were greater than sika deer (Table 2). The increase in threshold also led to the transition trend of Hill number and Peilou’s evenness from significant difference to insignificant difference, indicating that the increase in the threshold would underestimate the diversity of species’ diet and the degree of species preference (Figure 2). In addition, cattle have obvious phylogenetic specificity (sesMPD < 0; *p* = 0.01; Table 2), and the sika deer diet showed higher phylogenetic diversity (sesMPD < 0; *p* > 0.05; Table 2).

### 3.4. The Extent of Dietary Overlaps between Sika Deer and Cattle

The results from Tiangen show that both the sika deer and the cattle eat mainly Rosaceae, Betulaceae, Sapindaceae, and Urticaceae, accounting for 75% and 69% of their diet, respectively. Tsingke’s results are similar to those of Tiangen, with the four families accounting for a total of 77% of the diet of sika deer and 75% of the diet of cattle. The results of both companies showed that although the amount of each diet differed between sika deer and cattle, their main diets overlapped significantly. We compared diet composition among species using adonis, a permutational (PERMANOVA) that can accommodate both categorical and continuous predictor variables. To visualize patterns in dietary dissimilarity, we used nonmetric multidimensional scaling (NMDS). The diet composition of sika deer and cattle is significantly different (PERMANOVA: Tiangen, pseudo-*F*_1,70_ = 4.67, R^2^ = 0.06; *p* < 0.001; Figure 4A; Tsingke, pseudo-*F*_1,63_ = 4.00, R^2^ = 0.06; *p* < 0.01; Figure 4B). Interspecific dietary overlap using RRA metrics was measured by the Pianka index (*O_jk_*) and Bray–Curtis similarity (*BCsim* = 1 − Bray–Curtis dissimilarity), both ranging from 0 (complete divergence) to 1 (complete overlap;). Results from both companies showed significant overlap in the diets of sika deer and cattle (Tiangen: *O_jk_* = 0.834; *p* < 0.001; *BCsim* = 0.633; Tsingke: *O_jk_* = 0.845; *p* < 0.001; *BCsim* = 0.630; Figure 5). Indeed, the increase in the threshold seemed to have little effect on the niche overlap. With the increase of the threshold, the Pianka index increased first and then decreased slightly, while the *BCsim* index decreased slightly with the increase of the threshold, which had no significant effect on the results. (Figure 2)

## 4. Discussion

Although cattle are grazers in the traditional division of herbivores, in our study system their diet shifted to a predominance of dicotyledonous woody plants, and this diet shift resulted in a high degree of dietary overlap between sika deer and cattle. Moreover, compared to sika deer, cattle have higher dietary diversity and a wider dietary niche, implying that cattle are more competitive for forage plants. Our results show that overlapping dietary niches between cattle and sika deer may be the main reasons for the restricted distribution of the sika deer.

There are many factors that can affect the results when using high-throughput sequencing technology to analyze the diet, such as sequencing platforms [50] and data processing thresholds [45]. Different choices in these experiments and data processing may have different effects on the results. Therefore, in our study, to avoid the influence of experimental and processing errors on the results, we used two companies, multiple thresholds, and multiple indicators to calculate the dietary diversity and overlap between cattle and deer. Although there are some numerical differences in the results of these treatments, in general, these treatments have led to the consistent finding that the dietary diversity of cattle is significantly higher than that of deer, and the dietary overlap between the two is high.

Hofmann [32] classified grazers as species feeding almost exclusively on graminoids (less than 25% browse). Browsers (or concentrate selectors) feed on at least 75% of woody and nonwoody dicotyledonous plants, while mixed feeders feed on intermediate proportions of grass and browse. Cattle have traditionally been classified as typical grazer based on their digestion patterns, but in our study, we found that their feeding habits have undergone a great change. The grazer-browser diet shift in cattle or ‘cattle-type’ bovid species may not be rare. The main reasons for the shift in diet are food availability and nutritional requirements. Previous studies have found that some ‘cattle-type’ bovid species consume significant proportions of browse [34,51]. Pahl [33] also found that although cattle prefer monocotyledons, they also switch to dicotyledons when monocotyledons are scarce in the environment. Feeding types of cattle are probably determined by their habitats, such as cattle being grazers in open grassland, and browsers in dense woodland [52]. In addition to the availability of food resources, the nutritional content of the diet may also be a factor leading to dietary change. Craine et al. [53] found that climatic warming will increase the reliance of cattle on eudicots as protein concentrations of grasses decline. Seasonal variation of protein concentration also drives the grazer-browser diet shift in bison [54,55]. Our study area is a mixed conifer and broadleaf forest in the north, where dense trees and shrubs make the surface monocotyledons scarce. In addition, the nitrogen content in plants may be relatively low in this region [56]. Therefore, we speculated that the huge change in cattle eating habits in the local area may be caused by the joint effect of dietary resources and nutrition.

For herbivorous wild animals with similar ecological niches to domestic animals, interspecific competition between herbivores may cause niche partitioning and, as a consequence, a different use of resources [57,58]. Competition among ungulates is enhanced when the species are not coevolved, such as when one of them is an introduced species, including free-ranging domestic species [14,15,16,17,18]. The great dietary overlap between sika deer and cattle revealed in this study suggests strong competition between these two species in the National Park. We argue that introduced cattle have superior competitive abilities to sika deer. First, and most importantly, the difference in body size puts the sika deer at a relative disadvantage in competition. In addition, grazing cattle tend to act in clusters, and their numbers are much larger than those of wild sika deer populations [10], again making cattle more competitive. Secondly, the mobility of cattle is lower than sika deer [59]. The lower mobility of cattle may be due to their greater dietary diversity and wider dietary niche, which allows them to obtain ample food within a smaller range. Cattle feed on a wider variety of species, and they consume a high amount of food resources. This consumption of resources by cattle can reduce the forage quantity and quality of sympatric sika deer, which in turn limits the biomass of wild ungulate populations [28,29,30]. Therefore, bigger body size, lower mobility, and clustered cattle population may exclude sika deer and constrain their use of the National Park.

In our study area, sika deer have always been restricted to areas close to the border without expanding inland. Habitat selection by sika deer, obstruction by human roads, the introduction of electronic fencing, and competition for food caused by grazing could be among the factors contributing to this phenomenon. Sika deer may favor open habitats with more nutrient-rich food sources [60]. On the other hand, human roads and the installation of electronic fencing may act as barriers to the movement of sika deer into the National Park [5]. Grazing behavior in border areas further constricts the range of sika deer, and habitat fragmentation limits the potential for species dispersal and settlement by creating barriers to normal dispersal and settlement processes [61,62]. Separation of resources is required when two species share the same ecological niche in the same area [63]. The dietary niche width has also been shown to influence the geographical distribution of species [64]. Our study demonstrates that there is a significant overlap between the dietary niches of cattle and sika deer, which implies that the less competitive sika deer will inevitably be forced to change its distribution range, moving away from the cattle’s distribution range and gathering in the border area. Therefore, we suggest that the primary factor impeding the spread of sika deer into the interior is the extreme overlap in ecological niches for feeding that exists between cattle and sika deer.

Through the use of camera trap data, previous studies have demonstrated how domestic livestock can have an impact on sika deer by reducing habitat use, increasing spatial avoidance, or changing the animals’ daily activity patterns [10]. Our study confirms the impact of domestic animals in terms of food habits and concludes that the significant food overlap between domestic animals and sika deer is a major factor impeding sika deer habitat expansion. With the aid of high-throughput sequencing, we can study dietary habits in greater detail and precision, which helps us measure the level of competition between wild and domestic herbivores to suggest practical conservation. We suggest that grazing practices should be prohibited in and around border areas, especially in the areas where sika deer are distributed, while in areas far from the border relatively conservative interventions can be made to confine livestock, thus reducing the impact on forests and wild herbivores. These measures have the potential to rapidly benefit wild herbivores that are competitively inhibited, but they will require effective systems and strict measures by the relevant authorities to promote the conservation of landscapes inhabited by wild herbivores and their predators.

## 5. Conclusions

Our results indicate that the feeding habits of cattle, an introduced species, have shifted from grazer to browser in mixed conifer and broad-leaved forests in northeast China. This resulted in a high degree of diet overlap between cattle and a local major wild herbivore, the sika deer. The superior competitive abilities of cattle due to their bigger body size and more diverse diet composition restrict the sika deer from spreading to the inland of China, which may restrict the distribution of their main predators Amur tiger and leopard around the Sino-Russia border. Hence, to restore the biodiversity in the national park, prohibiting cattle grazing in the park is necessary and urgent.

## Figures and Tables

**Figure 1 animals-13-00561-f001:**
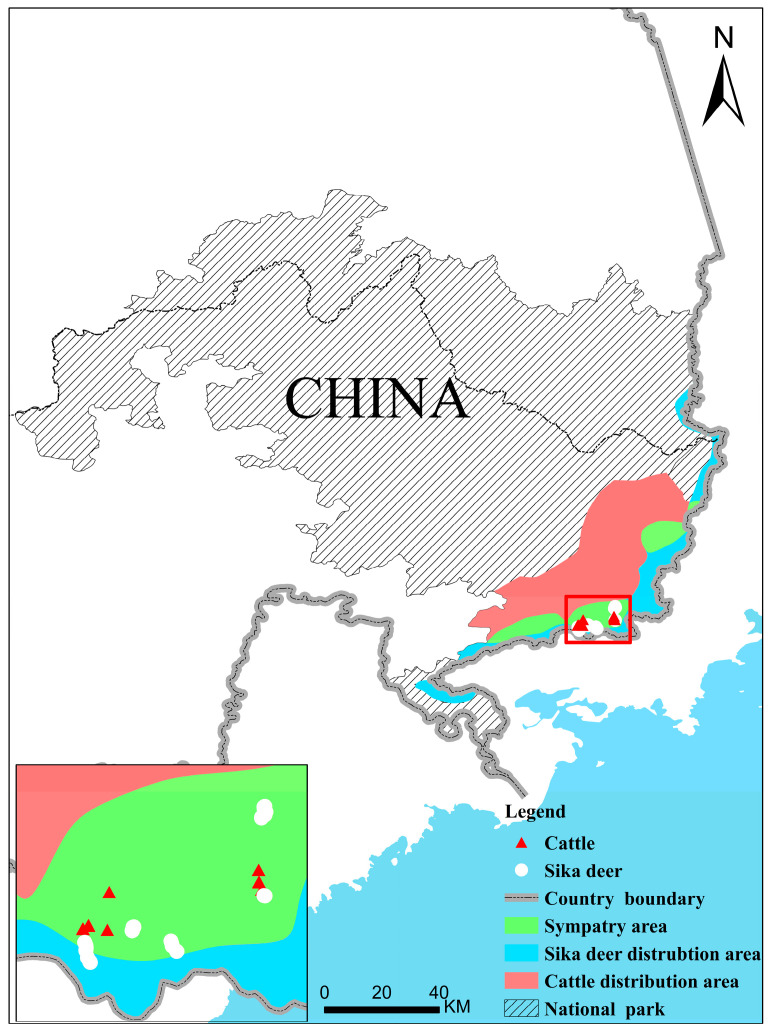
Distribution diagram of sampling points, data of the respective distribution areas, and sympatric distribution areas of cattle and sika deer are from Wang et al. [9].

**Figure 2 animals-13-00561-f002:**
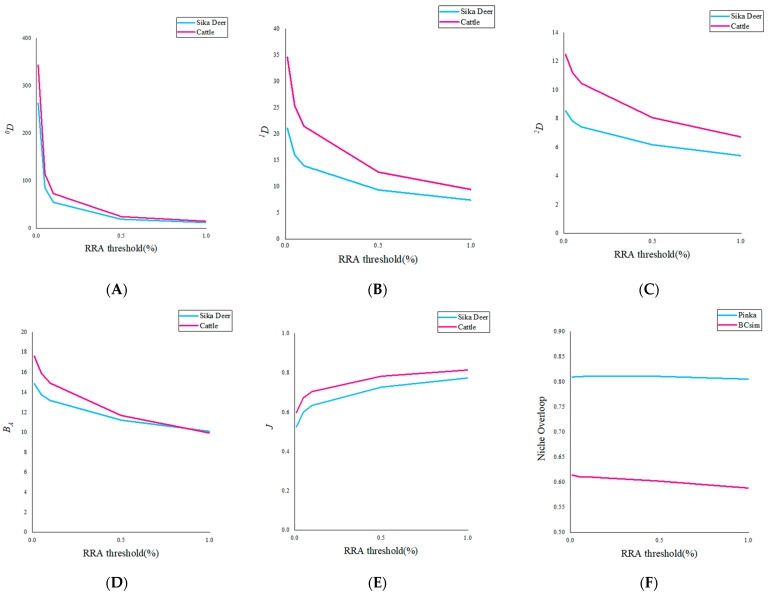
(**A**–**C**) ^0^*D*, ^1^*D*, ^2^*D* change with the increase of threshold. (**D**,**E**) *B_A_* and *J* change with the increase of threshold. (**F**) Two indices of niche overlap of cattle and sika deer varied with thresholds.

**Figure 3 animals-13-00561-f003:**
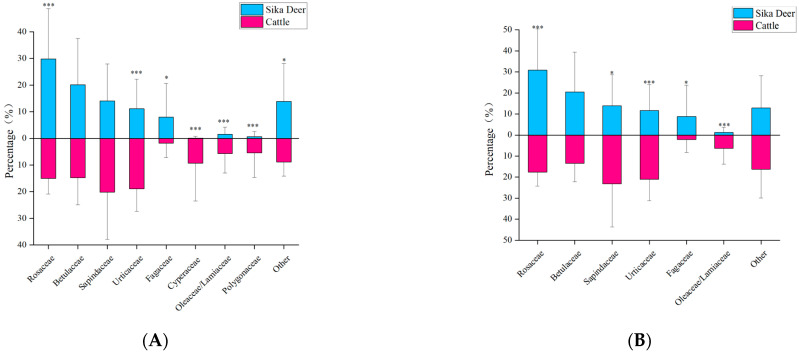
(**A**)The main food in the diet of cattle and sika deer from Tiangen. (**B**) The main food in the diet of cattle and sika deer from Tsingke. Families with less than 5% dietary share were classified as “Other”. Significance between two species of the same family was compared and statistically significant differences are shown. *, 0.01 < *p* < 0.05; ***, *p* < 0.01.

**Figure 4 animals-13-00561-f004:**
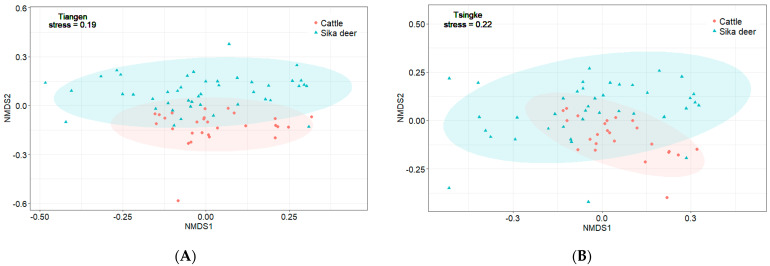
(**A**) NMDS of RRA-based Bray–Curtis dissimilarity of samples from Tiangen (stress = 0.19, PERMANOVA pseudo-*F*_1,70_ = 4.67, R^2^ = 0.06, *p* < 0.001), (**B**) NMDS of RRA-based Bray–Curtis dissimilarity of samples from Tsingke (stress = 0.22, PERMANOVA pseudo-*F*_1,63_ = 4.00, R^2^ = 0.06, *p* < 0.01).

**Figure 5 animals-13-00561-f005:**
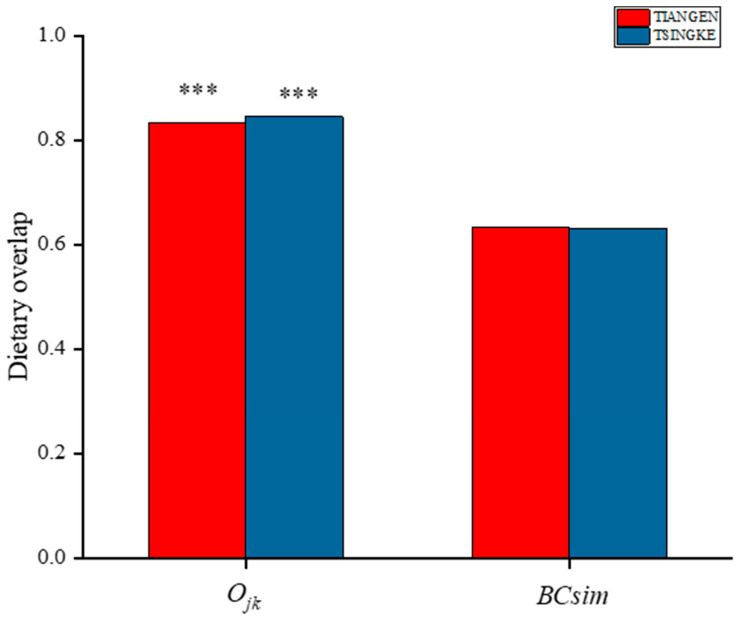
Pianka index (*O_jk_*) and Bray–Curtis similarity index (*BCsim*) of dietary compositions (RRA) between cattle and sika deer are shown. Statistically significant dietary overlaps estimated by *O_jk_* are shown. ***, *p* < 0.01. The different colors indicate the two companies.

**Table 1 animals-13-00561-t001:** Accounting of all sequences produced by next-generation sequencing.

Measure/Company	Tiangen		Tsingke	
	Count	Percentage	Count	Percentage
Raw sequence	13,422,242	100.0%	10,377,258	100.0%
Remove quality score < 40	13,252,796	98.7%	10,307,227	99.3%
matched tags and primers	11,971,284	89.2%	6,038,578	58.2%
Remove Sequences < 80 bp or total count < 10	11,522,651	85.8%	5,802,415	55.9%
Remove PCR and sequencing errors	7,300,952	54.4%	3,736,399	36.0%

**Table 2 animals-13-00561-t002:** Dietary taxonomic diversity is measured by Hill numbers of order number q = 0 (^0^*D*; species richness), q = 1 (^1^*D*; diversity and evenness), and q = 2 (^2^*D*; evenness giving more weight to rare species), and phylogenetic diversity measured by the mean pairwise phylogenetic distance (MPD) and mean nearest taxon distance (MNTD) using the RRA data of prey MOTUs are shown for each carnivore species from the three study areas. Statistically significant (*p* < 0.05) of phylogenetic diversity metrics are shown in bold. Two classic metrics for measuring dietary diversity, Levin’s niche breath (*B_A_*) and Peilou’s evenness (*J*), are also presented.

		Taxonomic Diversity					Phylogenetic Diversity					
	Species	^0^ *D*	^1^ *D*	^2^ *D*	*J*	*B_A_*	MPD	sesMPD	MPD *p*-Value	MNTD	sesMNTD	MNTD *p*-Value
Tiangen	Cattle	9	**6.07**	**4.94**	**0.82**	7.24	173.81	−2.71	**0.01**	88.61	0.89	0.80
	Sika deer	8	**4.87**	**3.85**	**0.75**	5.87	202.11	−0.83	0.23	103.37	1.26	0.88
Tsingke	Cattle	8	**5.6**	**4.55**	**0.81**	6.39	170.01	−2.56	**0.01**	90.78	0.98	0.83
	Sika deer	7	**4.34**	**3.45**	**0.72**	5.56	190.28	−1.08	0.15	118.81	1.84	0.94

## Data Availability

The data presented in this study are openly available in NCBI at https://www.ncbi.nlm.nih.gov/sra/PRJNA916368 accessed on 12 January 2023, reference number PRJNA916368.

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
