# Peer review of "Domestic Cattle in a National Park Restricting the Sika Deer Due to Diet Overlap"

_animals, 2023, doi:10.3390/ani13040561_

Round 1

Reviewer 1 Report

The paper investigates the diets to native sika deer and cattle livestock within the Northeast Tiger and Leopard National Park in China, to assess the extent of dietary overlap and hence competition between the two species. The authors use DNA meta-barcoding of plant DNA from faecal samples and primarily compare diet using relative read abundance. Samples from the two species were collected within relatively close proximity to minimise potential habitat effects, although the two species rarely occur in close proximity, which is good to see. The authors also compare the results from different sequence companies to assess if sequencing protocols impacted on the results.

The paper is well written and methods appear appropriate and the results are interesting. The shift in the typical grazing diet of cattle, when in environments where grasses are rarer is an important consideration when cattle occur in areas intended for conservation. I only have a few minor suggestions.

Minor suggestions:

I think the conclusion implied in the title is perhaps an over statement of the results, since the diet overlap alone may not be the only factor preventing invasion of the park by deer, as the authors indicate in the discussion. The dietary overlap is potentially also restricting the population in over ways. I also don’t think dispersal is quite the correct term here. Perhaps dispersal could be removed, or replaced with “presence of”.

Ln 81: this sentence starts talking about herbivores in general, I wonder if that last point about newborn deer could be generalised? Also, food resources is mentioned above so the following sentence seems unnecessary.

182: The thresholds that were tested could be explained in more detail.

Results – There is no mention of the results of negative controls. Please clarify that these produced no sequence data, or no relevant contamination, or if reads from relevant species were present, then how this was accounted for (e.g., removal of a certain number of reads in potentially impacted samples).

Fig 1: A box showing where on the larger map the insert represents would be helpful

References: There are a couple of reference that need the formatting corrected (i.e., titles all in capitals).

Reviewer 2 Report

Review for animals-2163165

This manuscripts presents an interesting analysis of the diet of a domestic and a wild ungulate using DNA barcoding, demonstrating that the domestic species (cattle) is a generalist and the wild species (sika) is a specialist. This is important because only the wild species is a prey for the endangered large predators of the study area. Thus, a critical ecosystem service is at stake.

I strongly suggest to rewrite the abstract completely. 1) The English is not very good. For example, in the first sentence: change to “Managers need to know the extent of conflict (…)”, change “greater competitiveness” into “superior competitive abilities”, etc. 2) Several important key words such as DNA barcoding, niche partition, dietary overlap, prey management for the conservation of large predators, are missing.

As I am not familiar with genetic analysis of diets, I did not understand the implication of the %RRA threshold. This needs better justification. This part of the analysis was not introduced in the abstract, so I wonder what role it plays.

Somewhere in the introduction and methods, you should insist on the fact that you did not need to analyse the availability of the different orders of plants because samples from the two species were collected in nearby places and therefore the availability was the same for both species.

The results that feral cattle shift to a browser diet is not new. Cattle do that in almost all instances where they are released in closed habitat. I strongly suggest to perform a review of the literature.

In several places of the manuscript, the authors imply that cattle are a superior competitor. However they do not bring any formal, objective evidence. Even if I believe the statement is true just because of the difference in size, the authors need to explain why they consider cattle as the superior competitor over sika deer.
